# Analysis of Protein Sequence Identity, Binding Sites, and 3D Structures Identifies Eight Pollen Species and Ten Fruit Species with High Risk of Cross-Reactive Allergies

**DOI:** 10.3390/genes13081464

**Published:** 2022-08-17

**Authors:** Wei Zhou, Kaylah Bias, Dylan Lenczewski-Jowers, Jiliah Henderson, Victor Cupp, Anthony Ananga, Joel Winyo Ochieng, Violeta Tsolova

**Affiliations:** 1Food Science Program, College of Agriculture and Food Sciences, Florida A&M University, 1740 S. Martin Luther King Jr. Blvd. Room 305-A Perry Paige South, Tallahassee, FL 32307, USA; 2Center for Viticulture and Small Fruits Research, College of Agriculture and Food Sciences, Florida A&M University, 6505 Mahan Drive, Tallahassee, FL 32317, USA; 3Agricultural Biotechnology Programme, University of Nairobi, P.O. Box 29053, Nairobi 00625, Kenya

**Keywords:** fruit allergen, cross-reactive, 3D structure, conserved domains, drug design

## Abstract

Fruit allergens are proteins from fruits or pollen that cause allergy in humans, an increasing food safety concern worldwide. With the globalization of food trade and changing lifestyles and dietary habits, characterization and identification of these allergens are urgently needed to inform public awareness, diagnosis and treatment of allergies, drug design, as well as food standards and regulations. This study conducted a phylogenetic reconstruction and protein clustering among 60 fruit and pollen allergens from 19 species, and analyzed the clusters, in silico, for cross-reactivity (IgE), 3D protein structure prediction, transmembrane and signal peptides, and conserved domains and motifs. Herein, we wanted to predict the likelihood of their interaction with antibodies, as well as cross-reactivity between the many allergens derived from the same protein families, as the potential for cross-reactivity complicates the management of fruit allergies. Phylogenetic analysis classified the allergens into four clusters. The first cluster (*n* = 9) comprising pollen allergens showed a high risk of cross-reactivity between eight allergens, with Bet v1 conserved domain, but lacked a transmembrane helix and signal peptide. The second (*n* = 10) cluster similarly suggested a high risk of cross-reactivity among allergens, with Prolifin conserved domain. However, the group lacked a transmembrane helix and signal peptide. The third (*n* = 13) and fourth (*n* = 29) clusters comprised allergens with significant sequence diversity, predicted low risk of cross-reactivity, and showed both a transmembrane helix and signal peptide. These results are critical for treatment and drug design that mostly use transmembrane proteins as targets. The prediction of high risk of cross-reactivity indicates that it may be possible to design a generic drug that will be effective against the wide range of allergens. Therefore, in the past, we may have avoided the array of fruit species if one was allergic to any one member of the cluster.

## 1. Introduction

Food allergy is an increasing food safety concern worldwide. Allergic reactions to food range from mild cases to fatalities, with more than 100 deaths occurring per year in the United States alone [1,2]. Severe cases where symptoms cause a life-threatening reaction called anaphylaxis are common, with symptoms that include difficulty breathing and swallowing, vomiting and diarrhea, dizziness, dangerously low blood pressure, swelling of the lips, tongue, throat, and other parts of the body, and loss of consciousness. Anaphylaxis usually occurs within minutes, but can occur up to several hours after coming into contact with/eating certain foods or their products. Allergy to fruits and pollen appear to be at the center of current food allergy research [3], as more people are reported to react to allergens (proteins from fruits or pollen that cause allergy in humans) in common fruits, such as apples, peaches, kiwi, cherries, grapes, strawberries, and bananas [4,5]. In some instances, even small amounts or contamination of food with certain fruits or pollen can cause a serious reaction, by inducing an allergic sensitization in susceptible individuals or by eliciting allergic reactions in those who are already sensitized.

Traditional drugs for managing food allergies work by blocking the interaction between the extracellular region of membrane proteins and ligands or inhibiting the activity of their intracellular regions. However, incomplete inhibition or induction of drug resistance remains a common problem with this approach [6,7,8]. Transmembrane proteins are involved in a broad range of biological processes; which explains why more than 50% of recently launched drugs target membrane proteins [9]. Furthermore, emerging protein-targeted degradation technologies, such as PROTAC, provide new insights for the design of anti-pollen allergy drugs. Other approaches currently used to alleviate the dangers of allergens from plant sources include (1) the removal of the offending protein from the food, an approach that depends on identifying the specific allergenic protein, then engineering the plant to not produce that particular protein, (2) empowering the body to lessen the allergic response, (3) altering the protein through genetic engineering/gene editing. As a result it is not recognized by the human’s immunoglobulin E (IgE antibodies) as the trigger for an allergic response, while the protein is still functioning normally. IgE are antibodies produced by the immune system that can trigger severe reactions to an allergen. The most common form is type I hypersensitivity (allergy) reactions, in which allergens cross-link high-affinity IgE receptors bind on the surface of basophils or mast cells, resulting in the release of local mediators, such as histamine [10,11,12]. On the other hand, IgE antibodies are usually found at the lowest concentration systemically as they become sequestered at cell surfaces through binding to high-affinity receptors [13,14]. The discovery of IgE has had a significant effect on the diagnosis and management of allergies, enabling clinicians to differentiate between IgE-mediated allergic diseases and other hypersensitivity reactions and to appropriately manage the IgE antibody-driven inflammation causing IgE-mediated allergic diseases [15].

One of the factors that still complicates the diagnosis and management of fruit allergies is cross-reactivity, which occurs when the proteins in one substance (such as pollen) are similar to the proteins found in another substance (a food). For example, food allergy to apples, hazelnuts, and celery is frequent in individuals with birch pollen (BP) allergy since IgE antibodies specific for the major birch pollen allergen, Bet v 1, cross-react with structurally related allergens in these foods, and T lymphocytes specific for Bet v 1 also cross-react with these dietary proteins [16,17]. This complicates the diagnosis and possibly explains why, for example, up to 20% of Americans have a perceived food allergy. However, the problem can be medically diagnosed in only about 2% of the population [18]. Excluding the disparity between statistics, confusion with food intolerance cannot be sufficiently attributed to this difficulty. With specific drugs still under development, the prospect of cross-reactivity, and the difficulty in totally avoiding these products, recent developments in genetic engineering offer a complementary chance to develop varieties with a significantly lower level of allergenicity, by removal of these proteins to deliver long-term health benefits and nutrition to millions of people who depend on or come into contact with these products.

This study aimed at the identification and in silico structural characterization of common fruit and pollen allergens. In addition, we herein investigated their possible interaction with antibodies and cross-reactivity between the many allergens derived from the same protein families, which potentially complicates the management of these allergies. To provide an essential stepping stone for drug development, genetic engineering, and consumer awareness, as well as dietary and behavioral considerations, the study analyzed cross-reactivity and possible drug targets.

## 2. Materials and Methods

### 2.1. Database Search and Sequence Retrieval

Allergen protein sequences of 19 species were collected from the World Health Organization and International Union of Immunological Societies (WHO/IUIS) Allergen database (http://allergen.org/ (accessed on 1 December 2021)). Based on approval by the World Health Organization and International Union of Immunological Societies (WHO/IUIS) Allergen Nomenclature Sub-committee, the website is the official site for the systematic allergen nomenclature. The WHO/IUIS Allergen Nomenclature Sub-Committee is responsible for maintaining and developing a unique, unambiguous, and systematic nomenclature for allergenic proteins. The nomenclature is based on the Linnean system and is applied to all allergens. A minimal criterion of demonstrated IgE binding to the suggested allergen using sera from patients allergic to the specific source is required.

### 2.2. Conserved Domain and Gene Family Analysis

For the identification of the number of domains in allergen protein, a domain search was executed by Conserved Domains Database (http://www.ncbi.nlm.nih.gov/Structure/cdd/cdd.shtml (accessed on 1 December 2021)) [19] and Pfam database (http://pfam.xfam.org/ (accessed on 1 December 2021)) [20] with both local and global search strategies. In addition, the Threshold of Expect Value is 0.01 and the Maximum number of hits is 500 amino acids in NCBI Conserved Domain Database. In Pfam database, all sequence regions that satisfy a family-specific curated threshold, also known as the gathering threshold, are aligned to the profile HMM to create the full alignment. Only a significant domain found in each protein sequence was considered as a valid domain.

### 2.3. Construction of the Phylogenetic Tree

Alignment and phylogenetic reconstructions were performed using the function “build” of ETE3 v3.1.1 [21] as implemented on the GenomeNet (https://www.genome.jp/tools/ete/ (accessed on 1 December 2021)). The tree was constructed using FastTree v2.1.8 with default parameters [22]. Values at nodes are SH-like local supports.

FastTree infers approximately-maximum-likelihood phylogenetic trees from alignments of nucleotide or protein sequences. FastTree uses the Jukes–Cantor or generalized time-reversible (GTR) models of nucleotide evolution and the JTT (Jones–Taylor–Thornton 1992) model [23], WAG [24] or LG models [25] of amino acid evolution. Accounting for the varying rates of evolution across sites, FastTree uses a single rate for each site (the “CAT” approximation) to estimate the reliability of each split in the tree. FastTree computes local support values with the Shimodaira–Hasegawa test (these are the same as PhyML 3.0 “SH-like local supports”).

### 2.4. Identification and Annotation of Conserved Motifs

The program MEME (v5.4.1) (https://meme-suite.org/meme/tools/meme (accessed on 1 December 2021)) [26] and ClustalW multiple alignment analyses (http://www.genome.jp/tools/clustalw (accessed on 1 December 2021)) were used for the elucidation of conserved motifs in all allergen protein sequences. The following parameters were used: Number of repetitions, any; maximum number of motifs, 8; and the optimum motif widths were constrained to between 6 and 60 residues. The logos of hidden Markov models (HMM) were used for the visualization of domain conservation [27].

### 2.5. Protein Structure Prediction

The three-dimensional (3D) protein structure prediction was performed using the Protein Homology/AnalogY Recognition Engine (Phyre2), an online protein fold recognition server (www.sbg.bio.ic.ac.uk/phyre2/ (accessed on 1 December 2021)) [28]. Phyre2 is a web-based service for protein structure prediction using the principles and techniques of homology modeling. It is able to regularly generate reliable protein models when other widely used methods, such as PSI-BLAST cannot.

### 2.6. Transmembrane and Signal Peptide Analysis

TMHMM 2.0 from DTU Health TCH was used for the prediction of transmembrane helices in proteins (https://services.healthtech.dtu.dk/service.php?TMHMM-2.0 (accessed on 1 December 2021)). TMHMM is a membrane protein topology prediction method based on a hidden Markov model. It predicts transmembrane helices and discriminates between soluble and membrane proteins with high degree of accuracy. In addition, TMHMM can discriminate between soluble and membrane proteins with both specificity and sensitivity better than 99%, although the accuracy drops when signal peptides are present. This high degree of accuracy allowed the prediction of reliably integral membrane proteins in a large collection of genomes.

### 2.7. Signal Peptide Analysis and Transmembrane Topology Prediction

Signal peptide analysis was performed using SignalP 6.0 (https://services.healthtech.dtu.dk/service.php?SignalP-3.0 (accessed on 1 December 2021)). SignalP 6.0 server predicts the presence and location of signal peptide cleavage sites in amino acid sequences from different organisms: Gram-positive prokaryotes, Gram-negative prokaryotes, and eukaryotes. The method incorporates a prediction of cleavage sites and a signal peptide/non-signal peptide prediction based on a combination of several artificial neural networks and hidden Markov models. In addition, SignalP 6.0 predicts the regions of signal peptides. Depending on the type, the positions of n-, h-, and c-regions as well as of other distinctive features are predicted. Moreover, to avoid misclassifying the membrane-spanning portion of a transmembrane protein and a membrane-spanning segment near the N-terminus in signal peptide, the signal peptide prediction and transmembrane topology prediction were performed simultaneously by software TMHMM, which was based on a combination of several artificial neural networks and hidden Markov models.

### 2.8. A-RISC Index Analysis

To estimate and visualize the likelihood of IgE cross-reactivity between allergens, we applied the A-RISC index (Allergens’–Relative Identity, Similarity, and Cross-Reactivity) in our study following the method proposed by Chruszcz et al. in 2018 [29]. The A-RISC index for a specific protein pair is defined as the average of protein sequence similarity (S) and identity (I). The index is a single numerical value that provides information on the relative homology between allergens from a particular protein family and selected members of that family. In the case of allergens, the physical meaning of the A-RISC index can be explained by considering the interaction of these proteins with antibodies. When comparing two protein sequences, the compared amino acids are divided into three groups: Identical, similar, and dissimilar. Similar is defined as: “similar” = “same” + “similar but not identical”. Only identical and similar amino acids are responsible for cross-reactivity. The following equation describes a subset of all amino acids that may interact with cross-reactive antibodies as follows:$$I+\frac{S-I}{2}=\frac{I+S}{2}$$
In addition, it provides the formula for calculation of the A-RISC index.

## 3. Results

### 3.1. Phylogenetic Clustering and Conserved Motifs

Sixty allergen protein sequences of 19 species were collected from the World Health Organization and International Union of Immunological Societies (WHO/IUIS) Allergen database. All of the 60 fruit allergens were designated to their Pfams (Appendix A).

Based on a combination of the phylogenetic and conserved motif analyses by MEME program, the 60 allergens were clustered into four phylogroups (clusters were marked as I, II, III, IV in Figure 1). Cluster 1 was separated from the fruit allergens at the root (Figure 1) and had high degree of intra-clade sequence similarity.

### 3.2. Cluster I

Cluster I (*n* = 9) comprised the following pollen allergens: Kiwi D8 (CAM31909.1), Apricot Ar1 (AAB97141.1), Raspberry I1 (XP_004296886.1), Raspberry_I1 (ABG54495.1), Strawberry A1 (ABG54495.1), Peach P1 (ABB78006.1), Cherry A1 (AAC02632.1), Apple D1 (CAA58646.1), and Pear C1 (AAC13315.1). This cluster showed a high risk of cross-reactivity (A-RISC index of 100%) between allergens from eight species, except for Kiwi D8 (Figure 2A).

Similarity of Kiwi D8 and Apricot Ar1 was 61% only, and for Kiwi D8 and the other allergens in Cluster I, it was lower than 60%, indicating a low risk of cross-reactivity between Kiwi D8 and other allergens (Figure 2A,B). Members had conserved motifs 4, 5, and 7. Whereas motif 4 was highly conserved, motifs 5 and 7 were absent in Raspberry1 and Kiwi D1, respectively (Figure 1). This cluster had Bet v 1 conserved domain (ligand-binding bet_v_1 domain of major pollen allergen of white birch (*Betula verrucosa*, Bet v 1), and belonged to START/RHO_α_C/PITP/Bet_v1/CoxG/CalC (SRPBCC) ligand-binding domain superfamily (Appendix A). Furthermore, the 3D structure of Cluster I allergens was characterized with two α-helices and five β-pleated sheets, except for Kiwi D8. The 3D structure of Kiwi D8 does not have the typical α-helix and β-pleated sheets as other allergens in its cluster (Figure 2C). We found no transmembrane helix and signal peptide motifs in Cluster I (Table 1). However, allergens were characterized with two α-helices and five β-pleated sheets.

### 3.3. Cluster II

Cluster II comprised the following 10 members: Pineapple C1 (AAK54835.1), Kiwi D9 (C0HL99.1), Strawberry A4 (XP_004287490.1), Melon M2 (AAW69549.1), Banana A1 (AAK54834.1), Apple D4 (AAD29414.1), Cherry A4 (AAD29411.1), Peach P4 (CAD37201.1), Date-Palm D2 (CAD10390.1), and Pear C4 (AAD29410.1).

Cluster II showed a high risk of cross-reactivity (A-RISC index) between allergens from 10 species (Figure 3A), with similarity of most allergens as higher than 76%. However, Kiwi D9 was 22 amino acids shorter than the other allergens in the cluster (Figure 3B). The 3D structure of most Cluster II allergens was characterized with four α-helices and three β-pleated sheets, except for Melon M2, which does not have the typical α-helix and β-pleated sheets as other allergens in the cluster (Figure 3C). Here, members had conserved motifs 2, 3, and 8, except for motif 2 that was absent in Kiwi D9, and motifs 3 and 8 that were absent in Apple D4. Motifs 2 and 3 were associated with allergens in Cluster II (Figure 1).

Allergens in Cluster II had Profilin conserved domain and belonged to Profilin super family (Appendix A). This was the case with Cluster 1, and in Cluster II, we found no transmembrane helix and signal peptide motifs (Table 1). However, a majority of allergens in Cluster II were characterized with four α-helices and three β-pleated sheets (Figure 3C).

### 3.4. Cluster III

Cluster III comprised the following thirteen allergens: Kiwi D10 (GFZ19186.1), Banana A3 (XP_009396870.1), Peach P3 (XP_007206159.1), Apricot (ADR66947.1), Plum D3 (PQP98495.1), Cherry A3 (AAF26449.1), Pear C3 (AAF26451.1), Apple D3 (AAT80633.1), Strawberry A3 (CAC86258.1), Raspberry I3 (ABG54494.1), Grape V1 (AAO33394.1), and Pomegranate G1 (AHB19227.1). This cluster showed a low risk of cross-reactivity (A-RISC index) between allergens from 19 species, with the highest sequence similarity as 60% only (Figure 4A,B). Cluster III exhibited sequence diversity; while their 3D structure was characterized with five α-helices (Figure 4C). Members had non-specific lipid-transfer protein conserved domain, and belonged to AAI_LTSS (α-amylase inhibitors (AAI), lipid transfer (LT), and seed storage (SS) protein family (Appendix A)). In addition, they had conserved motifs 1 and 3, except for Pomegranate G1 and Banana A3, which were less conservative in motif 3 than other allergens (Figure 1). Seven of the members had predicted transmembrane helix.

### 3.5. Cluster IV

Cluster IV comprised the following twenty-nine fruit allergens: Kiwi D7 (PSR89527.1), Peach P9 (XP_007199020.1), Peach P10 (M5X316), Kiwi D6 (BAC54964.1), Kiwi D12 (ABB77213.1), Pear C5 (AAC24001.1), Green Kiwi D5 (P84527.2), Apricot Ar5 (AAD32205.1), Melon M1 (BAA06905.1), Kiwi D11 (P85524.1), Green Kiwi D4 (AAR92223.1), Papaya P1 (ACV85695.1), Melon M3 (XP_008455060.1), Banana A2 (CAC81811.1), Pers A1 (CAB01591.1), Banana A5 (AAB82772.2_banana), Coconut N1 (ALQ56981.1), Cherry Av7 (XP_021820299.1), Peach P7 (XP_016648029.1), Green Kiwi D2 (CAI38795.2), Banana A4 (XP_009406737.1), Cherry A2 (AAB38064.1), Apple D2 (AAC36740.1), Peach P2 (ACE80959.1), Pineapple C2 (BAA21849.1), Papaya P2 (CAA66378.1), Green Kiwi D1 (CAA34486.1), Date M1 (AAX40948.1), and Pomegranate G14 (G1UH28). Cluster IV A-RISC indices showed a low risk of cross-reactivity between allergens from 10 species except between Banana A2 and Pers A1 (0.76); Cherry Av7 and Peach P7 (0.97), Green Kiwi D2 and Banana A4 (0.72); Cherry A2 and Apple D2 (0.7); Cherry A2 and Peach P2 (0.67); Apple D2 and Peach P2 (0.79) (Figure 5). Allergens in Cluster IV were more diverse (Appendix A) and had no significant conserved motif in any of the 29 allergens (Figure 1). Twelve allergens in this cluster had transmembrane helix (Table 1). The diversity of Cluster IV might be highlighted as a putative cluster that requires reiteration in the future, as more samples are recruited to the tree. New clusters could emerge from Cluster IV as more samples reveal new motifs that are conserved or shared between the elements.

### 3.6. Transmembrane Topology and Signal Peptide Prediction

To avoid misclassifying the membrane-spanning portion of a transmembrane protein and a membrane-spanning segment near the N-terminus in signal peptide, the signal peptide and transmembrane topology predictions were performed simultaneously by the software TMHMM, which was based on a combination of several artificial neural networks and hidden Markov models.

Based on our analysis, no transmembrane helix and signal peptide motifs were found in Clusters I and II (Table 1). However, Cluster I allergens were characterized with two α-helices and five β-pleated sheets, and most of the allergens in Cluster II were characterized with four α-helices and three β-pleated sheets (Figure 2C and Figure 3C).

Based on TMHMM prediction, all of the allergens in Cluster III were with signal peptides, and the probability is as high as 100%. Among them, peach_P3 (XP_007206159.1), apricot_Ar3 (ADR66947.1), plum_D3 (PQP98495.1), cherry_A3 (AAF26449.1), pear_C3 (AAF26451.1), strawberry_A3 (CAC86258.1), and raspberry_I3 (ABG54494.1) had one predicted transmembrane helix (Table 1).

The TMHMM prediction showed that, except for pear_C5 (AAC24001.1), apricot_Ar5 (AAD32205.1), and kiwi_D11 (P85524.1), all of the remaining 26 allergens were predicted with signal peptide. Twelve allergens in Cluster IV were predicted with transmembrane helix, including peach_P9 (XP_007199020.1), kiwi_D6 (BAC54964.1), greenkiwi_D4 (AAR92223.1), papaya_P1 (ACV85695.1), melon_M3 (XP_008455060.1), coconut_N1 (ALQ56981.1), greenkiwi_D2 (CAI38795.2), banana_A4 (XP_009406737.1), cherry_A2 (AAB38064.1), peach_P2 (ACE80959.1), pineapple_C2 (BAA21849.1), and greenkiwi_D1 (CAA34486.1).

## 4. Discussion

In this work, we studied 60 allergens from common fruits and classified them into four clusters. Each characterized by a specific level of sequence similarity, structural properties making them suitable drug targets or otherwise. 

These properties have implications for consumer choices, drug design, accelerated breeding for safer foods, as well as diagnostics and treatment/management.

### 4.1. Consumer Choices, Diagnostics, and Treatment

An increasing number of people across the world are allergic to food, particularly fruits and pollen. For example, about 20% of Americans have a perceived food allergy, but the problem can be medically diagnosed in only about 2% of the population [30]. The diagnosis and management could be complicated by, among other factors, cross-reactivity, since IgE antibodies specific for a major pollen allergen cross-react with structurally related allergens in some fruits and dietary proteins [16,17]. This study has predicted the likelihood of cross-reactivity between allergens derived from the same protein families in 19 common fruit species to better manage fruit allergies. The allergens were classified into four (4) clusters. Two of these clusters (I and II; results), showed high risk of cross-reactivity, suggesting that persons allergic to any one pollen or fruit of a cluster will most probably be allergic to other pollen/fruits within that cluster. Allergic patients must be aware of how their foods are prepared, and constantly manage the potential consequences of ingesting cross-contaminated foods. As plant allergens are most common allergens and are difficult to avoid, their identification and characterization are critical for the diagnosis and treatment of food allergies. The discovery of IgE has had a significant effect, enabling clinicians to differentiate between IgE-mediated allergic diseases and other hypersensitivity reactions and to appropriately handle the IgE antibody-driven inflammation causing IgE-mediated allergic diseases [15].

### 4.2. Implications for Drug Design

The mechanism by which allergens cause harm is now well understood, and this has provided opportunities for drug development in recent years. For example, it is known that many allergenic proteins are stable and slow to digest in the stomach. Rather than being quickly destroyed by digestion as most proteins, allergenic proteins remain intact longer, giving them time to prompt the allergenic response. This understanding of digestive stability in allergens has provided an opportunity to break down the chemical bonds in allergenic proteins by treating milk with thioredoxin H, a common, non-allergenic protein, rendering the milk safer. This approach is currently applied to wheat, soy, and other allergenic foods using genetic engineering to help break down the allergens. Traditional drugs are designed to block the interaction between the extracellular region of membrane proteins and ligands or to inhibit the activity of their intracellular regions. However, there are often problems of incomplete inhibition or induction of drug resistance [6,7,8]. Emerging protein-targeted degradation technologies, such as PROTAC, provide a new way of thinking of drug development, which directly eliminates the protein machinery that causes abnormal phenotypes by specifically degrading intracellular targets. This unique mechanism of action has greatly expanded the scope of drugs [31,32] and provides new leads for the design of anti-pollen allergy drugs. In addition to transmembrane proteins, signal peptides can be targeted. Many current and potential drug targets are membrane-bound or secreted proteins expressed and transported through the Sec61 secretory pathway. They target translocon channels across the endoplasmic reticulum (ER) membrane via a signal peptide (SP), a temporary structure at the N-terminus of their nascent chain [33]. During translation, these proteins enter the endoplasmic reticulum lumen and membrane through co-translational translocation. Small molecules have been found to interfere with this process, reducing protein expression by recognizing the unique structure of specific protein SPs. Therefore, SP may be an effective target for designing drugs for a variety of diseases, including some hereditary diseases [34,35,36,37].

This study showed that all allergens in Clusters I (pollen) and II lacked transmembrane helix and signal peptides, making drug design difficult. However, the prediction of high risk of cross-reactivity in these two groups indicates that it may be possible to design a generic drug that will be effective against the wide range of allergens. The findings of most allergens in Clusters III and IV with transmembrane protein and signal peptides (Table 1) suggest that medicines can be designed to target signal peptides for these allergens.

### 4.3. Implications for Genetic Engineering

Approach that is currently used to alleviate the dangers of allergens from plant sources includes the removal of the offending protein from the food. This approach depends on identifying the specific allergenic protein, then engineering the plant to not produce that particular protein and altering the protein through genetic engineering/gene editing. As a result, it is not recognized by the human’s IgE antibodies as the trigger for an allergic response, while the protein is still functioning normally. Recent developments in gene editing technologies offer a chance to develop varieties with a significantly lower level of allergenicity, by removal of these proteins to deliver long-term health benefits and nutrition to millions, depending on fruits and/or coming into contact with pollen or their products. This study provides structural details and an essential stepping stone for genetic engineering and gene editing for safer foods.

## 5. Conclusions

In this research, protein sequences of 60 plant allergens from common fruits were collected, classified, and analyzed. To provide and explore biological information to the greatest extent possible, multiple analyses were applied. This included phylogenetic analysis, motif analysis, protein 3D analysis, high risk of cross-reactivity analysis, and transmembrane and signal peptide analysis. To achieve more accurate gene clustering results, we combined the phylogenetic and protein motif analyses. Our investigation showed that these 60 proteins can be classified into four clusters and the motif classification matches the phylogenetic clusters very well. Moreover, we noticed that all pollen allergens were classified into Cluster I. Furthermore, for a better understanding of these allergens, we introduced the A-RISC and transmembrane and signal peptide analyses into the cross-reactivity analysis and found that Clusters I and II have high risk of cross-reactivity. These combinations provide a new direction for exploring the cross-reactivity of allergens. Our results are critical for treatment and drug design, which mostly use transmembrane proteins as targets. In the next steps, biochemical and biological experiments, which target the conserved domain and motif in Clusters I and II should be carried out first. A further confirmation of the cross-reactivity among allergens in Clusters I and II is also necessary.

Ultimately, of course, we have to acknowledge that even the bioinformatic research provides a type of guidance for laboratory practice. Therefore, our results still require verification by the immunological and allergological experiments and could be improved when more allergen information is available.

## Figures and Tables

**Figure 1 genes-13-01464-f001:**
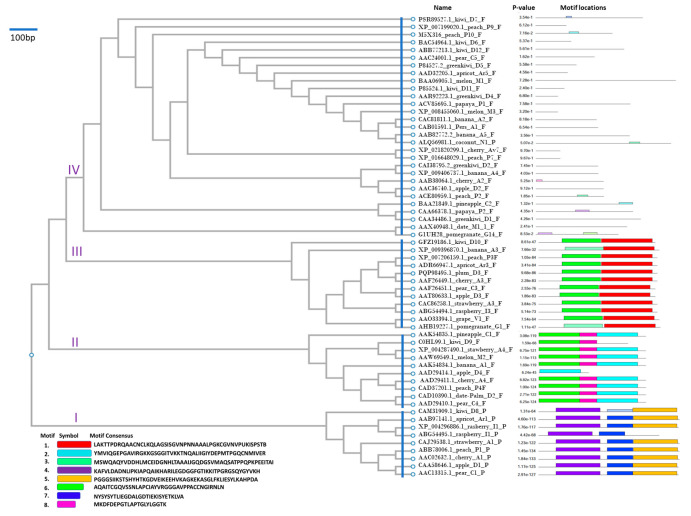
Phylogenetic and motif analyses. All of the 60 allergens were classified into four clusters, which were marked as I, II, III, IV. The logos of hidden Markov models and *p*-value of conserved motif for each allergen are attached.

**Figure 2 genes-13-01464-f002:**
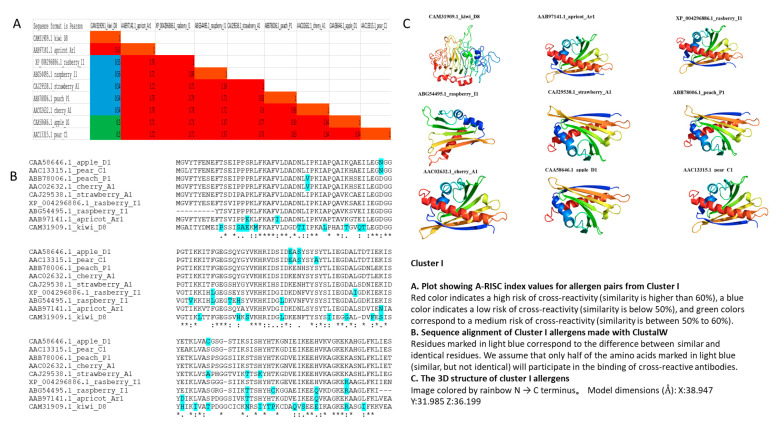
A-RISC index values, sequence alignment, and 3D structure of Cluster I.

**Figure 3 genes-13-01464-f003:**
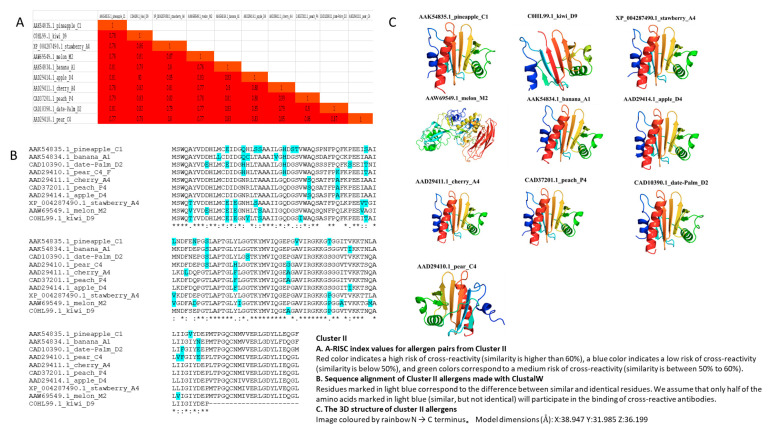
A-RISC index values, sequence alignment, and 3D structure of Cluster II.

**Figure 4 genes-13-01464-f004:**
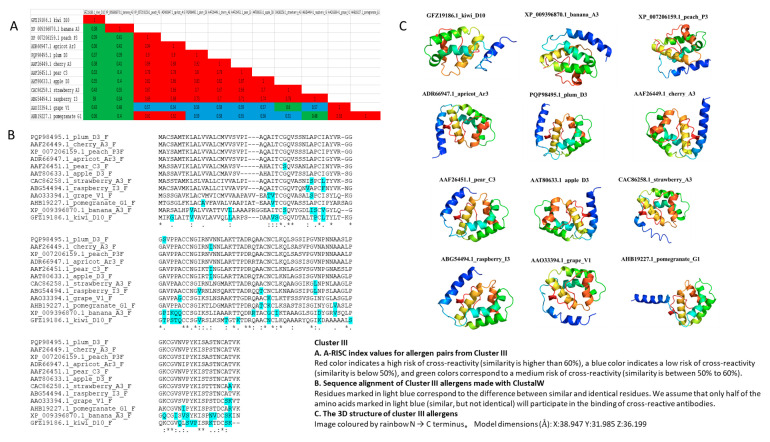
A-RISC index values, sequence alignment, and 3D structure of Cluster III.

**Figure 5 genes-13-01464-f005:**
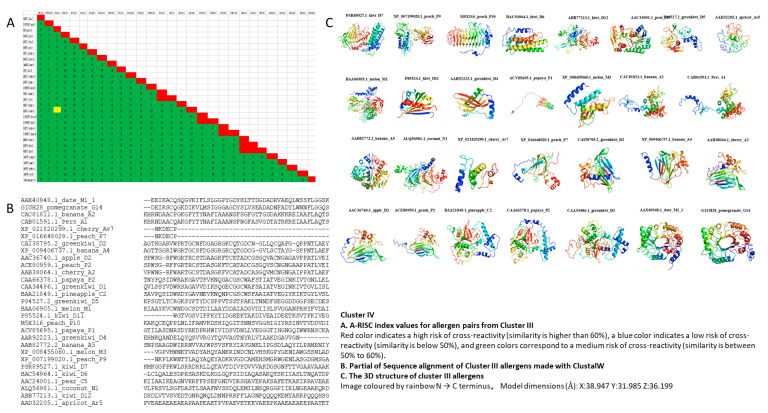
A-RISC index values, sequence alignment, and 3D structure of Cluster IV.

**Table 1 genes-13-01464-t001:** Transmembrane and signal peptide analysis results.

Cluster	Protein	Length	Number of Predicted TMHs	Exp Number of AAs in TMHs	Exp Number, First 60 AAs	Total Prob of N-in	Signal Length	Signal Peptide	Signal Peptide Probability	Signal Anchor Probability
Cluster I	CAM31909.1_kiwi_D8	157	0	0	0	0.14	70	no	0	0
Cluster I	AAB97141.1_apricot_Ar1	160	0	0	0	0.06	70	no	0	0
Cluster I	XP_004296886.1_rasberry_I1	159	0	0	0	0.04	70	no	0	0
Cluster I	ABG54495.1_raspberry_I1	137	0	0	0	0.06	70	no	0	0
Cluster I	CAJ29538.1_strawberry_A1	160	0	0	0	0.13	70	no	0	0
Cluster I	ABB78006.1_peach_P1	160	0	0	0	0.04	70	no	0	0
Cluster I	AAC02632.1_cherry_A1	160	0	0	0	0.04	70	no	0	0
Cluster I	CAA58646.1_apple_D1	159	0	0	0	0.06	70	no	0	0
Cluster I	AAC13315.1_pear_C1	159	0	0	0	0.04	70	no	0	0
Cluster II	AAK54835.1_pineapple_C1	131	0	0	0	0.3	70	no	0	0
Cluster II	sp|C0HL99.1_kiwi_D9	109	0	0	0	0.41	70	no	0	0
Cluster II	XP_004287490.1_stawberry_A4	131	0	0	0	0.35	70	no	0	0
Cluster II	AAW69549.1_melon_M2	131	0	0	0	0.25	70	no	0	0
Cluster II	AAK54834.1_banana_A1	131	0	0	0	0.3	70	no	0.01	0
Cluster II	AAD29414.1_apple_D4	61	0	0	0	0.25	70	no	0	0
Cluster II	AAD29411.1_cherry_A4	131	0	0	0	0.3	70	no	0	0
Cluster II	CAD37201.1_peach_P4	131	0	0	0	0.3	70	no	0	0
Cluster II	CAD10390.1_date-palm_D2	131	0	0	0	0.28	70	no	0.01	0
Cluster II	AAD29410.1_pear_C4	131	0	0	0	0.3	70	no	0	0
Cluster III	GFZ19186.1_kiwi_D10	115	0	1	1	0.22	70	yes	1	0
Cluster III	XP_009396870.1_banana_A3	119	0	13	13	0.37	70	yes	1	0
Cluster III	XP_007206159.1_peach_P3	117	1	17	17	0.8	70	yes	1	0
Cluster III	ADR66947.1_apricot_Ar3	117	1	17	17	0.8	70	yes	1	0
Cluster III	PQP98495.1_plum_D3	117	1	17	17	0.79	70	yes	1	0
Cluster III	AAF26449.1_cherry_A3	117	1	17	17	0.78	70	yes	1	0
Cluster III	AAF26451.1_pear_C3	115	1	18	18	0.52	70	yes	1	0
Cluster III	AAT80633.1_apple_D3	115	0	16	16	0.57	70	yes	1	0
Cluster III	CAC86258.1_strawberry_A3	117	1	21	21	0.8	70	yes	1	0
Cluster III	ABG54494.1_raspberry_I3	117	1	20	20	0.62	70	yes	1	0
Cluster III	AAO33394.1_grape_V1	119	0	15	15	0.53	70	yes	1	0
Cluster III	AHB19227.1_pomegranate_G1	120	0	11	11	0.44	70	yes	1	0
Cluster IV	PSR89527.1_kiwi_D7	559	0	10	10	0.46	70	yes	0.89	0.1
Cluster IV	XP_007199020.1_peach_P9	161	1	17	17	0.89	70	yes	0.99	0
Cluster IV	M5X16_peach_P10	401	0	13	13	0.54	70	yes	0.92	0.1
Cluster IV	BAC54964.1_kiwi_D6	185	1	18	18	0.83	70	yes	0.94	0.1
Cluster IV	ABB77213.1_kiwi_D12	462	0	0	0	0.01	70	yes	1	0
Cluster IV	AAC24001.1_pear_C5	308	0	0	0	0.05	70	no	0.01	0
Cluster IV	P84527.2_greenkiwi_D5	213	0	8	8	0.25	70	yes	1	0
Cluster IV	AAD32205.1_apricot_Ar5	168	0	0	0	0.1	70	no	0.01	0
Cluster IV	BAA06905.1_melon_M1	731	0	0	0	0	70	yes	0.97	0
Cluster IV	P85524.1_kiwi_D11_P	150	0	0	0	0.21	70	no	0	0
Cluster IV	AAR92223.1_greenkiwi_D4	116	1	16	16	0.29	70	yes	1	0
Cluster IV	ACV85695.1_papaya_P1	494	1	18	18	0.92	70	yes	0.52	0.2
Cluster IV	XP_008455060.1_melon_M3	116	1	21	21	0.85	70	yes	0.95	0
Cluster IV	CAC81811.1_banana_A2	318	0	0	0	0.02	70	yes	1	0
Cluster IV	CAB01591.1_pers_A1	326	0	15	14	0.66	70	yes	1	0
Cluster IV	AAB82772.2_banana_A5	340	0	14	13	0.66	70	yes	1	0
Cluster IV	ALQ56981.1_coconut_N1_P	490	1	20	19	0.97	70	yes	0.99	0
Cluster IV	XP_021820299.1_cherry_Av7	88	0	11	11	0.31	70	yes	1	0
Cluster IV	XP_016648029.1_peach_P7	88	0	11	11	0.31	70	yes	1	0
Cluster IV	CAI38795.2_greenkiwi_D2	225	1	19	19	0.85	70	yes	1	0
Cluster IV	XP_009406737.1_banana_A4	226	1	22	22	0.82	70	yes	1	0
Cluster IV	AAB38064.1_cherry_A2	245	1	21	21	0.92	70	yes	1	0
Cluster IV	AAC36740.1_apple_D2	245	0	0	0	0.1	70	yes	1	0
Cluster IV	ACE80959.1_peach_P2	246	1	18	18	0.77	70	yes	1	0
Cluster IV	BAA21849.1_pineapple_C2	351	1	16	16	0.88	70	yes	1	0
Cluster IV	CAA66378.1_papaya_P2	352	0	13	13	0.61	70	yes	0.99	0
Cluster IV	CAA34486.1_greenkiwi_D1	380	1	17	16	0.65	70	yes	0.95	0.1
Cluster IV	AAX40948.1_date_M1_1	330	0	4	4	0.21	70	yes	0.99	0
Cluster IV	G1UH28_pomegranate_G14	299	0	16	14	0.57	70	yes	1	0

Note: **Length:** The length of the protein sequence. **Number of predicted TMHs:** The number of predicted transmembrane helices. **Exp number of AAs in TMHs:** The expected number of amino acids in transmembrane helices. If this number is larger than 18, it is very likely to be a transmembrane protein (or have a signal peptide). **Exp number, first 60 AAs:** The expected number of amino acids in transmembrane helices in the first 60 amino acids of the protein. If this number is more than a few, you should be warned that a predicted transmembrane helix in the N-term could be a signal peptide. **Total prob of N-in:** The total probability that the N-term is on the cytoplasmic side of the membrane. **POSSIBLE N-term signal sequence:** A warning that is produced when “Exp number, first 60 AAs” is larger than 10.

## Data Availability

The data presented in this study are publicly available online.

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
