# Peer review of "Analysis of Protein Sequence Identity, Binding Sites, and 3D Structures Identifies Eight Pollen Species and Ten Fruit Species with High Risk of Cross-Reactive Allergies"

_genes, 2022, doi:10.3390/genes13081464_

Round 1

Reviewer 1 Report

This is an interesting article attempting to characterise protein allergens into four clusters and determining trends of cross-reactivity in each cluster. Overall, it is an interesting article that presents relevant data to the field. The results and methodology are presented clearly and are appropriate for the findings. The conclusion section elaborates in the direction of applicability of findings, which may border on too many postulations, and could be improved by instead discussing the trends and interpretation of the cladistic and molecular findings made in the results section. One major drawback of the conclusions herein that should be highlighted is that the sample size of the study is limited - only 60 allergens across 19 species were analysed, and it is questionable if enough statistical power is achieved when these are distributed into four clusters (no power calculation was given to affirm this, and no statistical test was given to demonstrate thresholds for significant amino acid differences required to cluster the sequences into unique clades). However, in context, this may yet be sufficient for a description of the data as presented. Specific comments on the article are given below:

p.1 l.37: In the abstract, the sample size should also be mentioned for third and fourth groups.

p.2 l.55: Citations should be provided for these examples: “… more people are reported to react to allergens (proteins from fruits or pollen that cause allergy in humans) in common fruits such as apples, peaches, kiwi, cherries, grapes, strawberries and bananas.”

p.2 l.89: Citations should be given for these statistics, “up to 20% of Americans have a perceived food allergy, but the problem can be medically diagnosed in only about 2% of the population”. Furthermore, since the former statistic is on ‘perceived food allergy,’ there are more factors to be considered in explaining the disparity between statistics, rather than simply the failure to identify definitively the allergic cause. This context is necessary to avoid misleading the reader into believing that it accounts for a major fraction of the disparity.

p.3 l.106: since the article references the WHO / IUIS Allergen Nomenclature database, a citation to the appropriate website should be given.

p.2 l.122: Further clarification is required here: “Only significant domain found in each protein sequence were considered as a valid domain.” What is considered a significant domain? Any output from the search meeting the 1.0 E value? Was there a manual curation performed? Were further filtering criteria performed?

p.3 l.141: The sentence structure here should be paraphrased to make it more readable “number of repetitions any, maximum number of motifs 8“. Additionally, why have the number of repetitions set to ‘any’ rather than a specific number? Without this knowledge, there is potential for a low number of iterations, which could bias the output.

p.3 l.143: “HMM Logos” should be capitalised and a reference made to the appropriate article(s).

p3. L147: reference to the publication(s) for PHYRE2 would be appropriate here, especially since authors are claiming it to be better for use here compared to PSI-BLAST.

Fig.1 has poor resolution, where the names and p-values cannot be read. Hopefully a high-res version is available for the final publication. The clades, 1- IV could be moved to the base of the earliest branch that includes the entire clade so that it is accurately depicting all elements in the clade, as well as for clarity of image. A scale for the axis in amino acid differences should also be provided.

Fig.2 - 5 also have poor resolution in the review version, which can hopefully be updated for the finalised submission. It is difficult to review these images in the current uploaded state.

p7. l.247: is there a reason why ‘significant’ is underlined here? Is it implying a statistical test? If not, perhaps it should be replaced with another term, such as ‘associated’ or ‘definitive of’ etc. From Fig.1, motif 2 is not unique to clade II (also present in clade I), and is not a defining feature of clade II (absent in kiwi D9). Similarly, motives 3 and 8 are absent in apple D4, so none of these characteristics can be said to be unique or definitive.

Table 1: why are some elements marked in red?

p.8 l.270: the elements in cluster IV seem to be very diverse, and the cluster itself seems to be more of a cluster defined by exclusion from the other clusters I – III rather than having defining features of its own. Perhaps this should be highlighted as a putative cluster that requires reiteration in the future, as more samples are recruited to the tree. Ostensibly, new clusters could emerge from cluster IV as more samples reveal new motifs conserved or shared between elements.

p.9 l.306: cluster IV is not capitalised. Sentence starts with a number, so it should be spelled out in full, ‘twelve’.

p.10 l.326: the allergens were grouped into four clusters. Word missing from the sentence. Furthermore, the reference to clusters in the result is spelled 1 and 11, where they should be I and II. Considering only clusters I and II show inter-clade cross-reactivity, can anything be concluded from the motifs present? For example, there is a shared motif 3 between clades II and III, but no motif 2 or 8 in clade III.

p.10 l.335: reference 12 is missing a closing bracket.

Author Response

Dear reviewer:

Thank you for your decision and constructive comments on our manuscript.

We totally agree and appreciate your comments about the statistical power of classification in phylogenetic analysis. This is a very common puzzle in phylogenetic analysis but is ignored by many authors. That is the reason our cluster analysis was not rely on Clustalw only. In this manuscript, to get a more accurate gene clustering result, we combined the phylogenetic analysis and protein motif analysis together. And our results showed that the motif analysis and Clustalw analysis are perfectly matched (Please find it in the Figure 1).  It double confirmed our clustering results is accurate and reliable.

We appreciate your suggestions on improving the conclusion section by discussing the trends and interpretation of the cladistic and molecular findings made in the results section. We have carefully made some changes and please check it with the attached word version manuscript with trackable changes.

We have tried our best to improve and made some other changes in the manuscript according to your comments. Please find the revision notes, point-to-point, are given as follows:

  1. Comments:1 l.37: In the abstract, the sample size should also be mentioned for third and fourth groups.

Response: Thank you for this suggestion, we have added the allergen number of third (n=13) and fourth (n = 29) cluster in the abstract section. Please check the changes in word version with trackable changes.

  1. Comments:2 l.55: Citations should be provided for these examples: “… more people are reported to react to allergens (proteins from fruits or pollen that cause allergy in humans) in common fruits such as apples, peaches, kiwi, cherries, grapes, strawberries and bananas.”

Response: Thank you very much for this suggestion. We have added the three citations in this part. They are:

Mastrorilli C, Cardinale F, Giannetti A, Caffarelli C. Pollen-food allergy syndrome: a not so rare disease in childhood. Medicina. 2019 Sep 26;55(10):641.

Muluk NB, Cingi C. Oral allergy syndrome. American journal of rhinology & allergy. 2018 Jan;32(1):27-30.

Please also find the changes in word version with trackable changes.

  1. Comments:2 l.89: Citations should be given for these statistics, “up to 20% of Americans have a perceived food allergy, but the problem can be medically diagnosed in only about 2% of the population”. Furthermore, since the former statistic is on ‘perceived food allergy,’ there are more factors to be considered in explaining the disparity between statistics, rather than simply the failure to identify definitively the allergic cause. This context is necessary to avoid misleading the reader into believing that it accounts for a major fraction of the disparity.

Response: Thank you very much for this suggestion. We have added the corresponding citations in this part.

Buchanan BB. Genetic engineering and the allergy issue. Plant Physiology. 2001 May;126(1):5-7.

Thanks for your further comments on food allergy statistics.  To make it clear, we revised the following sentence as : “Excluding the disparity between statistics, confusion with food intolerance cannot be sufficiently attributed to this difficulty.”

Please also find the corresponding changes in word version with trackable changes.

  1. Comments:3 l.106: since the article references the WHO / IUIS Allergen Nomenclature database, a citation to the appropriate website should be given.

Response: Thank you very much for this suggestion. We have attached the website of WHO / IUIS Allergen Nomenclature Home Page (http://allergen.org/) in this part.

Please find it in word version with trackable changes also.

  1. Comments:2 l.122: Further clarification is required here: “Only significant domain found in each protein sequence were considered as a valid domain.” What is considered a significant domain? Any output from the search meeting the 1.0 E value? Was there a manual curation performed? Were further filtering criteria performed?

Response: We appreciate these questions. We double checked the default settings of CD (conserved domain) search in NCBI database the Threshold of Expect Value is 0.01 and the Maximum number of hits is 500 amino acids. We revised it in the manuscript corresponding part. Yes, based on the searching software, the domain is considered as a significant domain only if the search meeting the threshold of E-value.  As Blast authors say in http://www.ncbi.nlm.nih.gov/BLAST/tutorial/Altschul-1.html, when E < 0.01, P-values and E-value are nearly identical. 10e-10 < E-value < 1 could be a true homologue but it is a gray area.

In domain searching part, we follow the default settings of software without any manual curation. We double checked the searching results in both of NCBI Conserved Domains Database (http://www.ncbi.nlm.nih.gov/Structure/cdd/cdd.shtml) and Pfam database (http://pfam.xfam.org/) for an accurate result.  In Pfam database, all sequence regions that satisfy a family-specific curated threshold, also known as the gathering threshold, are aligned to the profile HMM to create the full alignment. It is worth noting that a common misuse of Pfam is to use a single E-value threshold across all Pfam HMMs, which results in lower sensitivity and an increase in false positive matches when compared to using the per-family gathering thresholds.

Please find the corresponding changes in word version with trackable changes.

  1. Comments: 3 l.141: The sentence structure here should be paraphrased to make it more readable “number of repetitions any, maximum number of motifs 8“. Additionally, why have the number of repetitions set to ‘any’ rather than a specific number? Without this knowledge, there is potential for a low number of iterations, which could bias the output.

Response: Thank you very much for this comment. In MEME motif searching, we followed the default settings of MEME suite. At MEME website (https://meme-suite.org/meme/doc/meme.html), Any Number of Repetitions is explained as: “MEME assumes each sequence may contain any number of non-overlapping occurrences of each motif. This option is useful when you suspect that motifs repeat multiple times within a single sequence. In that case, the motifs found will be much more accurate than using one of the other options. This option can also be used to discover repeats within a single sequence. This option takes the much more computer time than the first option (about ten times as much) and is somewhat less sensitive to weak motifs which do not repeat within a single sequence than the other two options.”.

Thanks again for your great comments.

  1. Comments:3 l.143: “HMM Logos” should be capitalized and a reference made to the appropriate article(s).

Response: Thank you very much for this comment. We have changed it in the manuscript and the related reference is added.

Grundy WN, Bailey TL, Elkan CP, Baker ME. Meta-MEME: motif-based hidden Markov models of protein families. Bioinformatics. 1997 Aug 1;13(4):397-406.

  1. Comments: L147: reference to the publication(s) for PHYRE2 would be appropriate here, especially since authors are claiming it to be better for use here compared to PSI-BLAST.

Response: Thank you very much for this comment. We have added the related reference in the manuscript.

Kelley LA, Mezulis S, Yates CM, Wass MN, Sternberg MJ. The Phyre2 web portal for protein modeling, prediction and analysis. Nature protocols. 2015 Jun;10(6):845-58. Please find the corresponding changes in word version with trackable changes.

  1. Comments:1 has poor resolution, where the names and p-values cannot be read. Hopefully a high-res version is available for the final publication. The clades, 1- IV could be moved to the base of the earliest branch that includes the entire clade so that it is accurately depicting all elements in the clade, as well as for clarity of image. A scale for the axis in amino acid differences should also be provided.

Response: Thank you very much for this comment. The image resolution in word version is not high enough, but we provide the high-resolution figure separately as PNG file.  As you suggested, we have put the cluster label at the beginning of each cluster and it is looks much better now.  We also provide the 100bp amino acid scale bar. Please check it in word version manuscript.

  1. Comments:2 - 5 also have poor resolution in the review version, which can hopefully be updated for the finalized submission. It is difficult to review these images in the current uploaded state.

Response: Thank you very much for this comment. Same as figure 1, we also provide the PNG files for Figure 2-5. The resolution of these figures is good for publication.

  1. Comments: l.247: is there a reason why ‘significant’ is underlined here? Is it implying a statistical test? If not, perhaps it should be replaced with another term, such as ‘associated’ or ‘definitive of’ etc. From Fig.1, motif 2 is not unique to clade II (also present in clade I), and is not a defining feature of clade II (absent in kiwi D9). Similarly, motives 3 and 8 are absent in apple D4, so none of these characteristics can be said to be unique or definitive.

Response: Thank you very much for this comment. The underlined significant is a typo. We revised it to “associated to” now. Please check it. 

  1. Comments: Table 1: why are some elements marked in red?

Response: Thank you very much for this comment. It is a copy accident from the original analysis. We have changed the color to black.

  1. Comments: 8 l.270: the elements in cluster IV seem to be very diverse, and the cluster itself seems to be more of a cluster defined by exclusion from the other clusters I – III rather than having defining features of its own. Perhaps this should be highlighted as a putative cluster that requires reiteration in the future, as more samples are recruited to the tree. Ostensibly, new clusters could emerge from cluster IV as more samples reveal new motifs conserved or shared between elements.

Response: we appreciate allot for this comment. We added your comments “The diversity of Cluster IV might be highlighted as a putative cluster that requires reiteration in the future, as more samples are recruited to the tree. New clusters could emerge from cluster IV as more samples reveal new motifs conserved or shared between elements.” to part 3.5. Please find it in the word version with trackable changes.

  1. Comments: 9 l.306: cluster IV is not capitalized. Sentence starts with a number, so it should be spelled out in full, ‘twelve’.

Response: Thank you very much for this comment. We have changed these grammar errors. The “12” is replaced with “Twelve”. And “Cluster Iv” is changed to “Cluster IV”. Thanks again for your circumspection.  

  1. Comments: 10 l.326: the allergens were grouped into four clusters. Word missing from the sentence. Furthermore, the reference to clusters in the result is spelled 1 and 11, where they should be I and II. Considering only clusters I and II show inter-clade cross-reactivity, can anything be concluded from the motifs present? For example, there is a shared motif 3 between clades II and III, but no motif 2 or 8 in clade III.

Response: Thank you very much for this comment. We have changed the misspelling “1 and 11” to “I and II”.  It is a great idea to put the motif structure or conserved domain sequences into cross-reactivity analysis. Although the cross-reactivity analysis is based on sequence similarity, the motif structure and/or conserved domain would reduce the background noise and provide more accurate results. It would be a great research in future. Due to research limitations, we have to keep this research for next manuscript. We really appreciate your suggestions.

  1. Comments: 10 l.335: reference 12 is missing a closing bracket.

Response: Thank you very much for this comment. The missing closing bracket is added. 

Reviewer 2 Report

Analysis of protein sequence identity, binding sites and 3D structures identifies eight pollen species and ten fruit species with high-risk of cross-reactive allergies Worldwide, food allergens cause allergies in humans, an increasing food safety concern. Authors aimed to structurally characterize in-silico to identify common fruit and pollen allergens and to investigate their possible interaction with antibodies, and cross- reactivity between the many allergens derived from the same protein families, which potentially complicates the management of these allergies. Useful in drug development, genetic engineering, and consumer awareness as well as dietary and behavioural considerations, the study analyzed for cross-reactivity and possible drug targets. Well-written work with an adequate number of tables and figures. Conclusions are supported by the data, experimental work etc. The manuscript can be accepted for publication as it is.

Author Response

Dear reviewer:

Thank you for your decision and constructive comments on my manuscript. We totally agree and appreciate your comments. We have tried our best to improve and made some changes in the manuscript. In attached, please find the updated version of manuscript with trackable changes.

Best regards

Wei Zhou

Reviewer 3 Report

With interest, I read the manuscript genes-1839770-peer-review-v1. Overall, the aims of the study behind this work are justified and the study itself seems to be well designed and performed. The manuscript is in general nicely written.

There are, however, some inaccuracies and other problems that need to be attended.

Comments:

1.       The study is a fully in silico study. Lack of any wet work beyond the bioinformatics work should be mentioned as a limitation.

2.       In addition, the Authors should mention what the further experimental plans/steps would be. Besides, examples of the studies experimentally addressing such issues downstream (and/or upstream) of the in silico analyses should be given (PMID: 33805442).

3.       Please, modify the language a bit. E.g. line 73: “victim” or line “when our immune systems overreact in response to an allergy” (the immune system overreacts to antigens called in this case allergens and not to “allergy”. Etc.

4.       “IgE are antibodies produced when our immune systems overreact in response to an allergy. The antibodies attach to cells, releasing chemicals which then cause an allergic reaction [8, 9]. Otherwise they are typically found at the lowest concentration systemically as it becomes sequestered at cell surfaces through binding to its high-affinity receptor [10, 11].”. Please, be more exact. The allergic reaction including the release of “chemicals” form mast cells and basophils occurs upon cross-linking of the IgE-high affinity IgE receptors on those cells (PMID: 22909159).

5.       In general, please, verify the immunological and allergological part in very detail.

6.       “IgE” abbreviations is twice (lines 73 and 374). Please, check the abbreviations throughout the manuscript. Abbreviations appearing in the abstract should be independently explained there as well upon their first appearance.

7.       Figures should be much larger to make their details clearly visible. Either show them horizontally or change the design to vertical. Anyway, make the details 3-4 times bigger, please.

8.       Where are the Supplementary Materials? They are not visible to me in the submission system.

9.       Concluding paragraph should be added at the end of the Discussion.

Author Response

Dear reviewer:

Thank you for your decision and constructive comments on my manuscript. We totally agree and appreciate your comments. In attached, please find the updated version of manuscript with trackable changes. We have tried our best to improve and made some other changes in the manuscript according to your comments. Please find the revision notes, point-to-point, are given as follows:

  1. Comments:The study is a fully in silico study. Lack of any wet work beyond the bioinformatics work should be mentioned as a limitation. In addition, the Authors should mention what the further experimental plans/steps would be. Besides, examples of the studies experimentally addressing such issues downstream (and/or upstream) of the in-silico analyses should be given (PMID: 33805442).

Response: Thank you very much for this comment. We realize that the bioinformatic research could be a kind of guidance to laboratory practice, but the results from bioinformatics research still need to be confirmed with wet work.  As you suggested, we have added the conclusion part and addressed the limitation of bioinformatics research and proposed the experimental plans in next steps.  Please find it in the word version with trackable changes.

  1. Comments:Please, modify the language a bit. E.g. line 73: “victim” or line “when our immune systems overreact in response to an allergy” (the immune system overreacts to antigens called in this case allergens and not to “allergy”. Etc.

Response: Thank you very much for this comment. We have changed the “victim” to “human” in line 73 and line 361. We changed the “victim” to “we” in abstract, line 40.  We have changed the “allergy” to “allergen” in line 73.

We also carefully checked other typos and spelling errors in the manuscript and corrected them. Please find them in the word version with trackable changes.

  1. Comments: “IgE are antibodies produced when our immune systems overreact in response to an allergy. The antibodies attach to cells, releasing chemicals which then cause an allergic reaction [8, 9]. Otherwise they are typically found at the lowest concentration systemically as it becomes sequestered at cell surfaces through binding to its high-affinity receptor [10, 11].”. Please, be more exact. The allergic reaction including the release of “chemicals” form mast cells and basophils occurs upon cross-linking of the IgE-high affinity IgE receptors on those cells (PMID: 22909159).

Response: Thank you very much for this comment. The whole sentence is changed as “It is most common for their role in type I hypersensitivity (allergy) reactions. In type I hypersensitivity reactions, allergens cross-link IgE molecules and bind to high-affinity IgE receptors on the surface of basophils or mast cells, resulting in the release of local mediators such as histamine.”. and the related reference is added also.

Potaczek DP, Kabesch M. Current concepts of IgE regulation and impact of genetic determinants. Clinical & Experimental Allergy. 2012 Jun;42(6):852-71.

Please find the modification in the word version with trackable changes.

  1. Comments:In general, please, verify the immunological and allergological part in very detail.

Response: Thank you very much for this comment. We have tried our best to improve and made some changes in the manuscript. Please find the updated version of manuscript with trackable changes.

  1. Comments:“IgE” abbreviations is twice (lines 73 and 374). Please, check the abbreviations throughout the manuscript. Abbreviations appearing in the abstract should be independently explained there as well upon their first appearance.

Response: Thank you very much for this comment. We are sorry for this error. The whole sentence is changed as “…and, altering the protein through genetic engineering/gene editing so it is not recognized by the human’s IgE antibodies as the trigger for an allergic response, …”.

  1. Comments:Figures should be much larger to make their details clearly visible. Either show them horizontally or change the design to vertical. Anyway, make the details 3-4 times bigger, please.

Response: Thank you very much for this comment. We have modified the figure 1 as another reviewer’s suggestion. And for each figure, we separately provide the PNG files. The resolution of these figures is good for publication.

  1. Comments:Where are the Supplementary Materials? They are not visible to me in the submission system.

Response: Thank you very much for this comment. The supplementary materials are provided separately. We are sorry for not including in the review materials by system. We have attached it together with the updated manuscript. Please check it.

  1. Comments:Concluding paragraph should be added at the end of the Discussion.

Response: Thank you very much for this comment. Follow your suggestion, we have added a separate Conclusion part as follows, which including plans for next steps.

Conclusion: In this research, protein sequences of total 60 plant allergens from common fruits were collected, classified and analyzed.  To provide and explore biological information as much as we can, multiple analysis was applied which including phylogenetic analysis, motif analysis, protein 3D analysis, high-risk of cross-reactivity analysis, transmembrane and signal peptide analysis. To get a more accurate gene clustering results, we combined the phylogenetic analysis and protein motif analysis together. Our investigation showed that these 60 proteins can be grouped into four clusters and the motif classification matches the phylogenetic clusters very well.  We also noticed that all pollen allergens were grouped in cluster I.  To get a better understanding of these allergens, we also introduced the A-RISC analysis and transmembrane and signal peptide analysis into cross-reactivity analysis and found cluster I and cluster II have high-risk of cross-reactivity.  These methods combinations provide a new direction for exploring the cross-reactivity of allergens. Our results are critical for treatment and drug design that mostly use transmembrane proteins as targets. In next steps, biochemical and biological experiment which target to the conserved domain and motif in cluster I and II should be carried on first.  A further confirmation of the cross-reactivity among allergens in cluster I and II also is necessary. 

Of course, we have to acknowledge that even the bioinformatic research provide a kind of guidance for laboratory practice, our results still need to be verified by the immunological and allergological experiments and could be improved when more allergens information is available.”

Round 2

Reviewer 3 Report

Thank you very much for your work. Overall, my comments have been addressed well.

One small remark. Please, change:

“It is most common for their role in type I hypersensitivity (allergy) reactions. In type I hypersensitivity reactions, allergens cross-link IgE molecules and bind to high-affinity IgE receptors on the surface of basophils or mast cells, resulting in the release of local mediators such as histamine.”.

to:

“It is most common for their role in type I hypersensitivity (allergy) reactions. In type I hypersensitivity reactions, allergens cross-link the high-affinity IgE receptor-bound IgE molecules on the surface of basophils or mast cells, resulting in the release of local mediators such as histamine.”.